# Modified Manganese Phosphate Conversion Coating on Low-Carbon Steel

**DOI:** 10.3390/ma13061416

**Published:** 2020-03-20

**Authors:** Jakub Duszczyk, Katarzyna Siuzdak, Tomasz Klimczuk, Judyta Strychalska-Nowak, Adriana Zaleska-Medynska

**Affiliations:** 1Department of Environmental Technology, Faculty of Chemistry, University of Gdansk, Wita Stwosza str. 63, 80-308 Gdansk, Poland; 2The Szewalski Institute of Fluid-Flow Machinery Polish Academy of Sciences, Fiszera str. 14, 80-231 Gdansk, Poland; ksiuzdak@imp.gda.pl; 3Faculty of Applied Physics and Mathematics, Gdansk University of Technology, str. G. Narutowicza 11-12, 80-233 Gdansk, Poland; tomasz.klimczuk@pg.edu.pl (T.K.); judyta.strychalska@gmail.com (J.S.-N.)

**Keywords:** phosphating, corrosion, modified manganese coatings, manganese phosphating coatings, surface protection, conversion coatings, XRD

## Abstract

Conversion coatings are one of the primary types of galvanic coatings used to protect steel structures against corrosion. They are created through chemical reactions between the metal surface and the environment of the phosphating. This paper investigates the impact that the addition of new metal cations to the phosphating reaction environment has on the quality of the final coating. So far, standard phosphate coatings have contained only one primary element, such as zinc in the case of zinc coatings, or two elements, such as manganese and iron in the case of manganese coatings. The structural properties have been determined using a scanning electron microscope (SEM), X-ray diffraction (XRD), and electrochemical tests. New manganese coatings were produced through a reaction between the modified phosphating bath and the metal (Ba, Zn, Cd, Mo, Cu, Ce, Sr, and Ca). This change was noticeable in the structure of the produced manganese phosphate crystallites. A destructive effect of molybdenum and chromium was demonstrated. Microscopic analysis, XRD analysis and electrochemical tests suggest that the addition of new metal cations to the phosphating bath affects the corrosion resistance of the modified coating.

## 1. Introduction

Manganese phosphating is used to protect metal surface against corrosion. Metallic surfaces used in the industry are subject to wear and corrosion, which is why they have to be coated with a protective agent. There are many various types and processes of metal coating, which means that there are also many types of methods used for anticorrosion protection of metal. The methods include the conversion coating of metal with manganese phosphate [1,2,3,4,5,6,7]. During manganese phosphating, the steel product is submerged in a solution containing phosphoric (V) acid, nitric (V) acid and inorganic manganese and nickel compounds as well as organic phosphating accelerators containing NO_2_^−^ nitro groups, e.g., nitro derivatives of benzene or nitro-compounds derived from guanidine [8]. The quality of the final phosphate coating depends on the concentration of the phosphating bath, temperature, or duration of the process. Manganese conversion coatings increase the value and durability of the finished part, improving improving the adhesion between the metal substrate and the paint. Such coatings also protect against varnish corrosion. Manganese conversion coatings having a crystal structure. As a consequence, they have a very developed surface, which means they better absorb subsequent coatings. Phosphate coatings are widely used in practice. They provide corrosion protection for materials: during storage, at the transport stage, as excellent layers for other coatings (automotive and energy industries), as friction reducing material, as an insulator, or excellent material for oils and greases.

Manganese phosphate coatings are widely used coatings before treating them with another layer, which can be varnish, polymer, or oil. Crystals of di- and tri-substituted manganese phosphate increase the active surface of the metal, which increases the water absorption coefficient. These layers are used in the production of parts for refrigerators, bicycles, cars, and other devices, e.g., elevators. New phosphate coatings are created on steel with a graphite modified backing [9].

The phosphating process is improved through specific modifications, such as the addition of tartaric acid [10], copper or zinc [11], ethylenediamine [12], silicon compounds [13,14], and titanium compounds in order to improve the quality of the coating produced during the preparatory processes [15,16].

In this study, we used modifiers to introduce new chemical elements into the phosphating bath that could be used due to their availability or price. New elements in the manganese phosphate coating will enable a higher corrosion resistance of the phosphate coating. It was decided not to use compounds containing gold, silver, platinum, and rare-earth metals due to their high cost. The objective of this paper is to create a new phosphating bath that produces modified manganese coatings with higher corrosion resistance and improved integrity, which is why the study involved new inorganic chemical compounds introduced into the phosphating bath. 

## 2. Experimental

### 2.1. Sample Preparation

The plates made of low-carbon steel with a thickness of 0.8 mm and size of 20 mm × 20 mm were kept in toluene (analytical grade) before the tests to protect them against corrosion. Tests of the quality of the steel used in the study were carried out using the Solaris CCD Plus Optical Emission Spectrometer. The composition of steel used in the tests is given in Table 1. 

The samples were subsequently degreased in a 10% (*m/m*) solution of NaOH (Chempur, Poland) at 75 °C for 4 min, with an addition of 0.05% (*m/m*) of sodium dodecyl sulfate (SLS) (Sigma–Aldrich) as detergent. The samples degreased this way were washed with isopropyl alcohol and then with distilled water. After this process, the samples were etched in hydrochloric acid with a concentration of 15% (*m/m*) with an addition of 3% (*m/m*) H_2_O_2_ and 0.05 g of 1,2,3-benzotriazole (POCH, Gliwice, Poland). The process was carried out for 2 min at 21 °C. After this stage, the samples were washed several times with distilled water. After washing and etching as described above, the samples were kept under a layer of toluene in order to protect them against the corrosive effects of atmospheric air until they could be subject to the phosphating process. 

Immediately before phosphating, the samples were routinely activated in a solution containing equal content of manganese hydrogen phosphate MnHPO_4_ (Chemetall) and sodium diphosphate Na_4_P_2_O_7_ (Chemetall) in the amount of 0.2 g per 200 g of water. The process was carried out for 4 min at 45 °C. The solution was continuously stirred due prevent sedimentation process. Afterwards, without rinsing, the samples were subject to phosphating for 15 min at 97 °C with continuous stirring in accordance with the procedure specified in Table 2. 

### 2.2. Preparation of the Standard Phosphating Bath

In order to make the phosphating bath, it was necessary to prepare the manganese (II) phosphate (V) Mn_3_(PO_4_)_2_, which is not commercially available (Reaction (1)). To do so, a solution of phosphoric (V) acid with a concentration of 25% (*m/m*) (Sigma–Aldrich) was prepared, and it was mixed with metallic manganese (Sigma–Aldrich). Then, the entire compound was boiled at 60 °C until all of the metallic manganese was dissolved. After dissolving all of the added metallic manganese, the solution was slowly evaporated to dryness so as not to allow for the manganese phosphate to disintegrate (Reaction (1)). It was found that the phosphoric (V) acid had to be used in excess; otherwise, the reaction produced water-insoluble manganese hydrogen phosphate (Reaction (2)). To prevent this, phosphoric acid and manganese at a ratio of 4:1 were used. The chemical reactions used to produce two forms of manganese phosphate are given below:

with excess phosphoric (V) acid:2 H_3_PO_4_ + 3 Mn → Mn_3_(PO_4_)_2_ + 3 H_2_↑(1)
with deficiency of phosphoric (V) acid:H_3_PO_4_ + Mn → MnHPO_4_ + H_2_↑(2)

Manganese phosphate prepared as above was used to prepare the standard and modified phosphating baths. The typical composition of the phosphating bath is given in Table 3.

### 2.3. Preparation of Modified Phosphating Baths

The proposed qualitative composition of modified phosphating baths is given in Table 4. A more stable and safer derivative of nitroguanidine, i.e., 1-methyl-3-nitroguanidine, was used as the accelerator of the process. Soluble forms of compounds such as nitrates (V) or oxides, which readily react with the phosphating bath, were used to investigate the impact of the specified elements on the quality of the produced phosphate coating. Cerium (II) nitrate (V), barium nitrate (V), cadmium (II) oxide, zinc oxide, strontium (II) nitrate (V), calcium carbonate, copper (II) nitrate (V) and—due to the absence of molybdenum nitrate—sodium molybdate were used for this purpose.

### 2.4. Microstructure of the Manganese Coating

The specific characteristics of modified manganese coatings were determined with the use of scanning electron microscopy with EDS analysis (Energy Dispersive X-ray Spectroscopy). The quantitative analysis was carried out using the mapping method. The morphology of the produced phosphate coating was determined using a scanning electron microscope (FEI Company) equipped with an EDS attachment to enable analysis of the elemental composition of the coating. 

### 2.5. X-Ray Diffraction Analysis

X-ray diffraction (XRD) tests were carried out in order to determine the phase composition of the final coating and the size of produced crystallites. The measurements were performed on a Philips/PANalyticalX’Pert Pro MPD diffractometer with Cu K_α_ source (λ = 1.5404 Å). The data was collected from 2Θ = 5° to 70° with a scan speed of 1.1 deg/min at room temperature. The average crystallite size was estimated on the basis of the Scherrer equation taking into account reflections originating from (200) and (110) crystal phases and located at around 10°–11°.

### 2.6. Electrochemical Analysis

The objective of electrochemical tests was to determine the effect the additives have on the corrosion resistance of the manganese phosphate coating and to determine the protective properties of the phosphate coating present on the low-carbon steel. The electrochemical measurements were performed in a standard three-electrode arrangement using potentiostat–galvanostat PGStat302N. The sample of low-alloy steel after the particular modification was used as a working electrode (WE) with exposed surface area of 4 cm^2^. To verify the repeatability of the recorded data, 3 samples were prepared as WE for each materials and tested in fresh electrolyte. The reference electrode was Ag/AgCl/3M KCl (REF) and the Pt mesh, which was used as a counter electrode (CE). The electrochemical cell was filled with 150 mL of aerated 0.5 M KCl solution that was changed for a fresh one for each sample. The linear voltammetry (LV) scans were registered from −1.0 to −0.1 V vs. Ag/AgCl/3M KCl at the sweep rate of 1 mVs^−1^. The linear Tafel segments of the cathodic and anodic curves were extrapolated to the corrosion potential to obtain the corrosion current densities. The corrosion rate (CR) was calculated on the basis of equation: CR (mpy) = (0.13·J_corr_·E.W.)/d(3)
where J_corr_ is the density of the corrosion current (µA cm^−2^), E.W. is the equivalent weight and for low-alloy steel it equals 27.9 g, whereas d is the density of the sample material (7.87 g cm^−3^) [17,18]. The number 0.13 originates from the conversion factor with own unit: mpy/(µAcm) [19]. The error in the corrosion current density equals 0.01 µA cm^−2^, whereas in the case of corrosion rate it equals to 0.0046 mpy (mils per year).

### 2.7. Integrity of the Coating

The integrity of the coating is measured by the number of pores present in its structure per unit of area. In this study, the integrity of the produced coatings was tested in accordance with the procedure included in Polish Standard PN-81H-97016. The integrity of the coating was determined using the ferroxyl indicator. In accordance with the instructions included in the standard, the integrity of the coating is deemed to be good if there are no more than two colored points per 1 cm^2^ of the tested area [20].

### 2.8. Coating Mass

The purpose of the study was to determine the mass of the produced coatings and the impact of the modifier on their relative mass. Coating mass tests were carried out in accordance with the procedure specified in the European Standard EN ISO 3892:2001 European Standard. Coating mass was determined using a solution of chromium trioxide CrO_3_ [21].

## 3. Results and Discussion

This study investigated the impact of the addition of a modifying element on the quality of the produced coating, including the structure and nature of the created crystallites. SEM, XRD, and electrochemical tests confirmed the impact of the modifying additive on the quality of the final coating. A sample coated with the standard manganese phosphate coating was prepared during the study in addition to the modified samples based on the methods specified in Table 2. The sample prepared in this way was the reference sample for the other coatings produced from modified phosphating baths. The remaining phosphate samples were prepared in accordance with Table 4. 

On the basis of the tests of the cross-section of such coatings, it can be found that there are no two distinct border between the metal and the coating in the standard sample. The coating itself is produced from the substrate [8].

The results showed that the layer covering the metal surface had variable morphology, and the sizes of produced manganese phosphate crystallites significantly (most of them were several micrometers in length). 

### 3.1. Analysis of the Morphology of the Samples

Before phosphating the samples were prepared in accordance with the testing procedure specified in Table 2 and the modified phosphating baths had the composition indicated in Table 3 and Table 4. It was decided not to use substances such as chlorides, sulfates, and acetates during the preparation of the samples modified with chemical compounds that were highly soluble in water. It was observed that, when these substances had been used, the produced coating was damaged at rinsing stage after the phosphating process (data not shown). Two standard reference samples, i.e., the manganese phosphate sample and zinc phosphate sample were prepared at the phosphating line of Mayr Polska in order to compare the modified coatings with coatings used in the industry (Figure 1). 

In the prepared reference samples, there is a noticeable difference in the composition and structure of the produced coating. The zinc coating is made up of large crystals, with size ranging from 1 to 10 µm and the size of the manganese phosphate crystals ranges from 1 to 5 µm. Due to the regular needle shape of manganese phosphate crystallites, such coatings can easily be damaged even by a scratch. The zinc coating is not as sensitive to mechanical damage as the manganese coating due to the consistent shape of zinc phosphate crystals (Figure 1). 

The crystallinity of the obtained coatings is given in Table 5. The sample modified with barium has a structure similar to the sample modified with calcium. A different structure can be observed in the sample modified with cadmium, which is similar to the coating modified with strontium.

### 3.2. Impact of the Metal Additives on the Phosphating Bath

It was found that the modification of the phosphating bath with various inorganic chemicals had a considerable impact on the quality of the produced phosphate coating. The SEM analysis showed that the individual samples which were the subject of the phosphating process had a range of different structures (Figure 2). This means that the modifying additive had an impact on the quality of the produced coating. 

#### 3.2.1. Impact of the Addition of Molybdenum

The first additive introduced to the phosphating bath was sodium molybdate. The coating produced by the bath with sodium molybdate was not consistent and was easily damaged during rinsing in demineralized water. 

Ns_2_MoO_4_ (analytical grade) sodium molybdate was used in the investigation of the impact of the addition of cations on the produced coating. The coating modified by the molybdenum compound did not have a specific structure because it was a uniform layer without distinct individual crystals as in conversion coatings. There are many individual layers, creating a coating with only partial integrity. The coating does not have a uniform structure but, is instead it is made up of several superimposed layers. The primary component of the coating is iron (III) oxide, which can be observed in the EDS spectrum (Figure 3).

The coating prepared at the same time without the molybdenum compound has well-developed crystals (Figure 4a), and the coating produced with the addition of the molybdenum compound contains over 70% iron (Figure 4b). The use of the molybdenum compound did not produce a coating with good integrity. Molybdenum added to the phosphating bath is harmful, similarly to chromium (VI) compounds, which are also strongly oxidizing.

Molybdenum in the substrate (Figure 5) is also an example of the harmful effect of molybdenum. 42CrMoS4 + QT steel is most commonly used in the industry due to good technical properties, such as: abrasion resistance, hardness, and resistance to temperature. This steel is frequently used as a material for parts and components used in the automotive industry that are subject to high loads. It is also used to construct machinery [22].

Chromium-molybdenum steel has a wide range of applications due to good mechanical properties and suitability for heat treatment, including quenching and tempering, but the produced coating has a significantly inferior visual quality due to the size of the produced crystals. The coating produced on chromium-molybdenum steel is darker and less resistant to abrasion. The composition of chromium-molybdenum steel is specified in Table 6.

The results indicate that molybdenum included in the phosphating bath and in the substrate has a harmful effect on the produced phosphate coating.

Two phases can be observed in all of the tested samples: Fe–Ni and Mn–H–O–P. The Fe–Ni phase comes from the substrate material coated with the phosphate coating; this is the phase that contains iron (Fe), possibly with a small admixture of Ni (according to the crystallographic database, the maximum admixture of nickel in Fe is 30% Ni. Beyond this point, there is a change of structure, which has been observed in the sample modified with copper, which is why it has two Fe–Ni phases). There are no additional phases related to the addition of different metals to the bath, but SEM photos show that the shapes and sizes of the crystallites vary.

The coating produced in a phosphating bath with an addition of molybdenum has a light brown color (Figure 6). However, such a coating does not have any mechanical properties because it is easily destroyed even by touching.

#### 3.2.2. Impact of the Addition of Calcium

Calcium carbonate CaCO_3_ (analytical grade) (Chempur, Piekary Śląskie Poland) was also used as a modifier in this study. It was decided not to use calcium oxide and calcium hydroxide due to their strongly basic properties, which result in the immediate precipitation of calcium phosphate and increase the Ph of the phosphating bath, adversely affecting the quality of the produced coating. The phosphating process produced a coating with crystals in the form of thick and thin needles. The structure of this coating is very similar to the standard manganese coating. The produced crystals of the phosphate coating with the addition of calcium carbonate have regular shapes with a size of 1–5 μm. The sample modified with calcium has a structure similar to the structure of the sample modified with barium. The coating produced from the bath with the addition of calcium does not have any phase other than Mn-O-P-H (Figure 7). The crystallites of this coating have the shape of sharp-tipped prisms.

The modified sample produced from a bath with the addition of the calcium compound creates a powdery coating that does not meet all of the requirements for conversion coatings: i.e., uniform dark color, no visible streaks, and no powdering (Figure 8). 

#### 3.2.3. Impact of the Addition of Strontium

The addition of strontium to the phosphating bath caused the coating to have a different crystallite structure. The sample produced from a bath modified with strontium nitrate has multilayer lamellar crystallites with a regular, prism-like structure. The coating contains up to 1% strontium, which radically changes its properties. Due to the multilayer structure of the crystallites, the coating has strong absorbent properties owing to the large area (Figure 9).

Only the Mn-H-O-P phase can be observed in the coating modified with strontium (Figure 10).

#### 3.2.4. Impact of the Addition of Zinc

Another example is the use of zinc as a modifier of the manganese phosphating bath. Zinc has a wide range of applications, and there are many studies on the use of zinc to produce phosphate coatings. However, these studies involve higher concentrations of nickel and zinc relative to the manganese content, which is why the coatings produced this way should not be regarded as manganese phosphate coatings modified with zinc but as zinc phosphate coatings modified with nickel and manganese [18] or zinc coatings modified with manganese with an addition of copper [25] or nickel-free baths with an addition of zinc and copper [26]. The produced phosphate coating modified with zinc oxide has a non-specific structure. The crystals have irregular shapes (Figure 11), with large voids between them, causing the surface to retain substances, e.g., zirconium which was used during passivation, which is why it can be observed in the EDS spectrum (Figure 12). The difference can be observed by comparing SEM microphotographs of the zinc phosphate coating to manganese phosphate coating and manganese phosphate coating modified with zinc.

Introduction of zinc into the manganese coating improves its resistance to corrosion without changing its properties such as, for instance, its color. The manganese phosphate coating modified with zinc has large crystallites with large intercrystalline voids (Figure 11b). The coating modified with zinc has a single phase: Mn–H–O–P (Figure 12).

#### 3.2.5. Impact of the Addition of Copper

The sample produced from the bath modified with copper has a different structure. It is made up of spherical copper crystallites which grew manganese phosphate crystals (Figure 13). An interesting idea was to use copper (II) nitrate (V), which, unlike copper (II) oxide is highly soluble. The coating produced from the phosphating bath modified with copper nitrate has different properties, starting from the coppery color and ending with the corrosion resistance and rate of corrosion. The coating produced from the bath modified with the copper compound in accordance with the procedure specified in Table 2 and part of the Cu–Ni–Mn solution in Table 4 has two layers of irregularly-shaped crystals. The spherical precipitations form an additional outer layer made up of crystals containing primarily copper (Figure 13). 

The average size of the produced manganese phosphate crystals in the bath modified with copper is within 1 μm in the case of copper precipitations and within 5 μm in the case of the crystals of the substrate layer. XRD tests confirmed the presence of two Fe-Ni phases and a manganese phosphate phase. Due to its characteristic crystal structure (Figure 13 and Figure 14), the coating does not have suitable mechanical properties, such as scratch resistance. 

Crystals with an irregular shape that are covered with a large number of crystals with a shape resembling “balls” of copper are the reason why such precipitations can be easily torn off from the substrate. 

#### 3.2.6. Impact of the Addition of Cadmium

Another interesting example was the use of cadmium in a nickel and cadmium bath and a nickel-free cadmium bath. Phosphating with an admixture of cadmium and nickel produced a coating covered with regular uniform crystals in the form of sharp-tipped thick prisms with oval shapes. Most crystallites are within the range of 10 to 15 μm, with no voids between them. In some points, there are aggregates of small needles and ball-shaped crystals, whose size is from 1 to 5 μm, respectively (Figure 15b).

#### 3.2.7. Impact of the Addition of Barium

The sample modified with barium has a structure similar to the sample modified with calcium. A different structure can be observed in the sample modified with cadmium, which is similar to the coating modified with strontium. Crystallites formed from the phosphating bath modified with barium have distinct, sharp edges (Figure 16). The sample formed from the bath modified with barium has large crystallites with oval edges. 

The Mn-H-O-P phase can be observed in the sample. The EDS analysis shows a small fraction of barium in the phosphate coating (Figure 17).

#### 3.2.8. Impact of the Addition of Cerium

The sample modified with cerium has a structure similar to the sample modified with cadmium in the nickel-free configuration. Crystallites formed from the phosphating bath modified with cerium have distinct edges, and the crystallites have the shape of prisms (Figure 18). 

Preparation of the sample modified with cerium required a larger amount of the cerium compound due to the precipitation of the insoluble cerium phosphate from the bath, which necessitated frequent filtration of the phosphating bath during sample preparation. The Mn–H–O–P phase can be observed in the sample. The EDS analysis shows a small fraction of cerium in the phosphate coating (Figure 19).

### 3.3. Electrochemical Analyses

The test results indicate a significant impact of copper, strontium, cadmium and zinc compounds. Due to the high toxicity of cadmium compounds, the use of cadmium as a modifier of the phosphating bath may be problematic, even though cadmium is used in so-called marine coatings applied on ship components. Considering the toxicity of cadmium and its compounds and the fact that a large amount of wastewater would be generated during phosphating (approximately 15 m^3^/16 h), its industrial use would pose problems with suitable treatment of wastewater before its discharge into the sewer system. 

The results of electrochemical tests (Table 7) showed an impact of the new elements added to the phosphating bath (Figure 20).

Nevertheless, the lowest corrosion current and the corrosion rate that equal to 0.251 μAcm^−2^ and 0.11 mpy, respectively, was found for the Zn-Mayer-ref sample. The zinc oxide layer also enables efficient protection of the low carbon steel, taking into account both the value of the corrosion potential and current density. Other reference sample, labelled as Mn-Mayer-ref exhibits the highest corrosion current and finally the highest corrosion rate. Taking into account the morphology inspection, we believe that such significant difference between the performance of both reference materials results from their morphological features. Moreover, the shape of the anodic segment of Tafel plot differs a little from other ones presented in Figure 20 (right side) and some change in the slope of the anodic current is present in the range from −0.3 to −0.5 V. It could be related to the changes in the metal dissolution mechanism and adsorption of corrosion products to some active centers [27]. According to the SEM images analysis, Zn-reference is composed of consistent grains that makes the samples resistant against any hard damage comparing to Mn-reference built of needle-like structures. The high values of current density and corrosion rate was also found for the material modified with the copper containing bath. As was already revealed by SEM inspection, the coating does not exhibit proper mechanical properties and therefore cannot be regarded as appropriate against the corrosion since inclusions could be easily detached from the surface. Summarizing, the presence of different metal ions in the modification bath significantly affects the protection of the metal samples against the corrosion process while the most positive impact can be achieved using barium salt and zinc oxide coating. Moreover, the distinctive morphology features that were observed onto the SEM images, especially if surface is densely packed or some sharp edges are visible could improve or worsen the steel protection, respectively.

The results of the electrochemical test (Figure 20) for the steel sample that underwent treatment in Cu(NO_3_)_2_ solution enabled determination of very high corrosion rate of 21.9 (CR/mpy) (Table 7). It indicates that approach with copper ions containing bath induced highly lower resistance towards corrosion process. Another noteworthy element is strontium and zinc. The latter is a better modifier because its corrosion potential is −0.474 (Ecor/V), and its corrosion rate is only 0.37 (CR/mpy), which supports this conclusion considering the low current density of 0.811 (jcor/μAcm^−2^).

The data obtained in the electrochemical test (Table 7) shows that the coating modified with copper has the highest corrosion rate, reaching 21.9 (CR/mpy), with a corrosion potential of −0.504 (Ecor/V), and does not qualify as the best solution for corrosion protection of metal. The phosphate coating produced in the bath modified with copper nitrate has many precipitations, and the XRD tests indicate the presence of two phases: Fe-Ni phase from the surface of the substrate and Fe phase with a small admixture of Ni.

The next sample with the best electrochemical properties is the sample produced in the phosphating bath modified with barium (II) nitrate (V). The corrosion rate for this sample is 0.87 (CR/mpy), with current density of 1.88 (jcor/μAcm^−2^), and corrosion potential of −0.344 (Ecor/V). 

### 3.4. Integrity of the Coating

The samples prepared in accordance with the procedure stated in Table 2 and Table 4 were subject to an analysis of integrity. The test is found to be positive if blue spots appear on a strip of filter paper. If the number of spots per 1 cm^2^ is higher than two, it is deemed that the sample does not have integrity. 

In order to determine the integrity of the coating, 10 samples were prepared for each of the applied modified coatings, and they were subsequently tested. The test was performed using a ferroxyl solution consisting of sodium chloride and potassium hexacyanoferrate. A strip of filter paper previously soaked in the ferroxyl indicator for 20 s was placed on the metal sample with the phosphate coating. The standardized results of the integrity test are shown in Figure 21.

The results of the integrity analysis indicate that the sample modified with molybdenum does not meet the criteria for integrity, i.e., up to two colored points per 1 cm^2^. As indicated by SEM analyses, the sample is completely covered with iron oxide, which is why it does not meet any of the criteria for anticorrosion protection. The sample modified with the zinc compound has two colored spots on an area of 2 cm^2^, which means that this sample meets the integrity criterion. The sample modified with copper nitrate has seven colored spots on an area of 2 cm^2^, which means that the sample does not meet the criterion for coating integrity. The remaining samples meet the coating integrity criterion. 

### 3.5. Coating Mass

The samples prepared in accordance with the procedure stated in Table 2 and Table 4 were subject to an analysis of integrity. In order to determine the integrity of the coating, 10 samples were prepared for each of the applied modified coatings, and they were subsequently tested. This was done using the calibrated PA Pioneer scales series manufactured by OHAUS, type PA214CM/1.

The coated 4-cm^2^ metal sample with a known mass was weighed and then placed in a 15% (*m/m*) solution of chromium (VI) oxide for 15 min. The process was conducted at 75 °C. After this time, the sample was rinsed, dried and weighed again. The mass of the sample was determined using the following formula: *m*_A_ = [(*m*_1_ – *m*_2_)/A]∙10 (g/m^2^)(4)
where:*m*_1_—mass in milligrams of the coated test sample;*m*_2_—mass in milligrams of the test sample after coating dissolution;A—area of the coated test sample in square centimeters (8 cm^2^).

The results of the mass measurement are given in Table 8.

The results of the mass measurement indicate that the reference sample covered with the zinc coating has the lowest coating mass, and the sample modified with strontium has the highest mass. Consequently, the sample produced in the phosphating bath modified with strontium has the highest mass per 1 square meter. The higher the coating mass per unit of area, the heavier the final component. Based on the coating mass, it can be found that the modifier added to the phosphating bath affects the mass of the produced coating. 

## 4. Conclusions

(1)The study found that the molybdenum compound had a harmful impact on the quality of the created phosphate coating.(2)Based on a comparison of the electrochemical results for the standard manganese sample, zinc sample, and modified manganese sample, it can be found that the addition of zinc to the manganese bath has a significant impact on corrosion resistance. The corrosion rate for the standard manganese sample is 7.58 CR/mpy, and for the manganese sample modified with zinc 0.37 CR/mpy.(3)The results of the integrity analysis indicate that the sample modified with molybdenum does not meet the criteria for integrity. The sample modified with the zinc compound has two colored spots on an area of 2 cm^2^, which means that this sample meets the integrity criterion. The sample modified with copper nitrate has six colored spots on an area of 2 cm^2^, which means that the sample does meet not the criterion for coating integrity. The remaining samples meet the coating integrity criterion.(4)The largest coating mass was recorded for the sample modified with barium (0.030 g/m^2^), and the smallest mass was observed in the sample modified with cerium (0.013 g/m^2^).(5)The chemical composition of the metallic substrate has a significant influence on the produced phosphate coating. The coating produced on chromium-molybdenum steel had worse visual quality and lower abrasion resistance.(6)Summarizing the results of SEM tests, electrochemical tests and XRD tests and measurements of integrity and coating mass, it was found that the best anti-corrosive properties were demonstrated by the phosphate manganese coating modified with zinc. The corrosion rate for the manganese phosphate coating is 7.58 CR/mpy, and for the coating modified with zinc 0.37 CR/mpy. Comparison of the mass of the manganese phosphate coating (0.010 g/m^2^) and the manganese coating modified with zinc (0.014 g/m^2^) indicated that the latter coating has the best protective properties.

## Figures and Tables

**Figure 1 materials-13-01416-f001:**
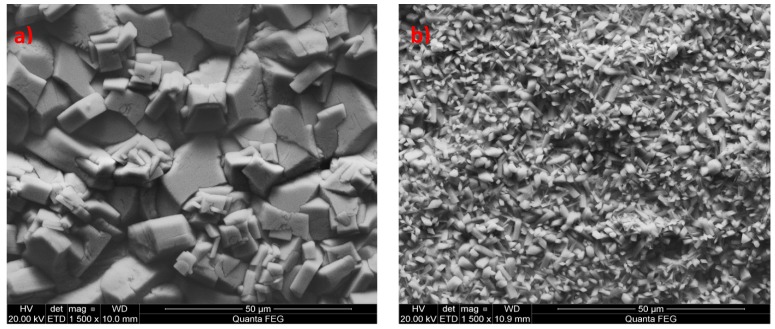
Standard reference coatings. scanning electron microscope (SEM) image with EDS: (**a**) zinc phosphate coating, (**b**) manganese phosphate coating.

**Figure 2 materials-13-01416-f002:**
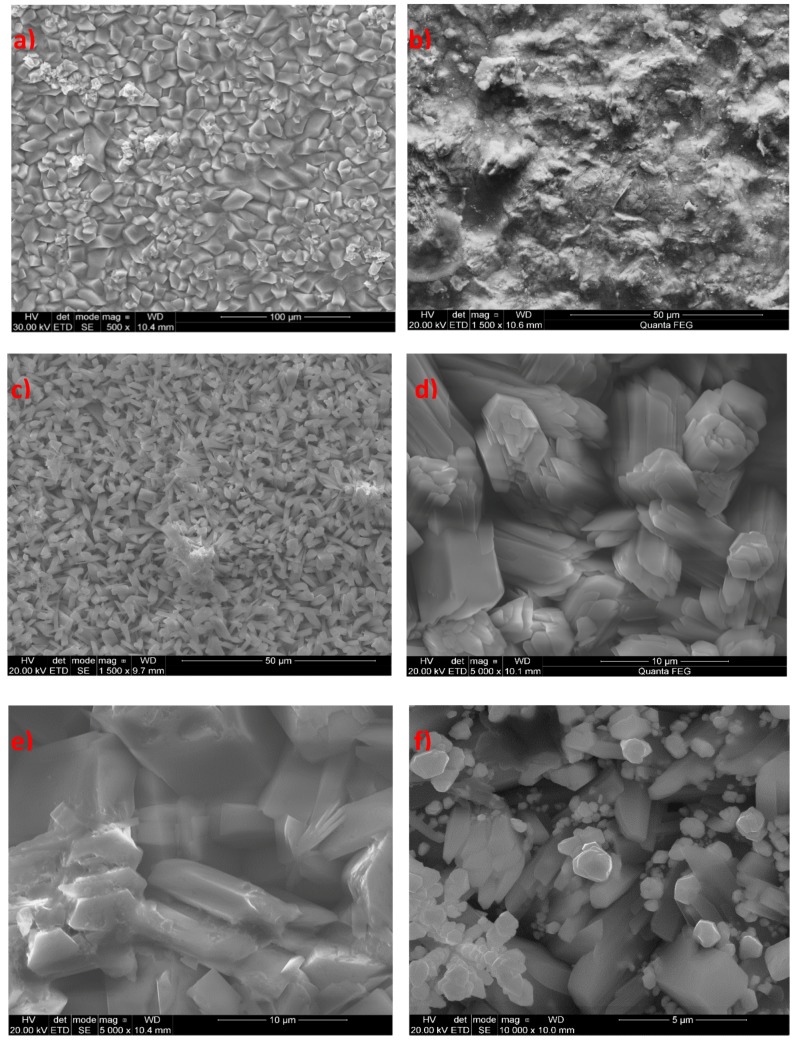
SEM image of the manganese phosphate coating: (**a**) modified with: Na_2_MoO_4,_ 1500× magnification (**b**), CaCO_3_, 1500× magnification (**c**), Sr(NO_3_)_2_, 5000× magnification (**d**), ZnO, 5000× magnification (**e**), Cu(NO_3_)_2_, 10,000× magnification (**f**)**,** Ba(NO_3_)_2_, 1500× magnification (**g**), CdO, 1500× magnification (**h**), and Ce(NO_3_)_2_, 1500× magnification (**i**).

**Figure 3 materials-13-01416-f003:**
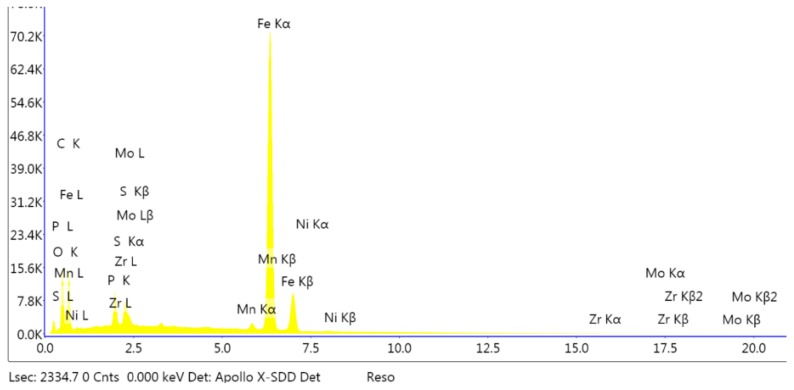
EDS spectrum of the sample modified with molybdenum.

**Figure 4 materials-13-01416-f004:**
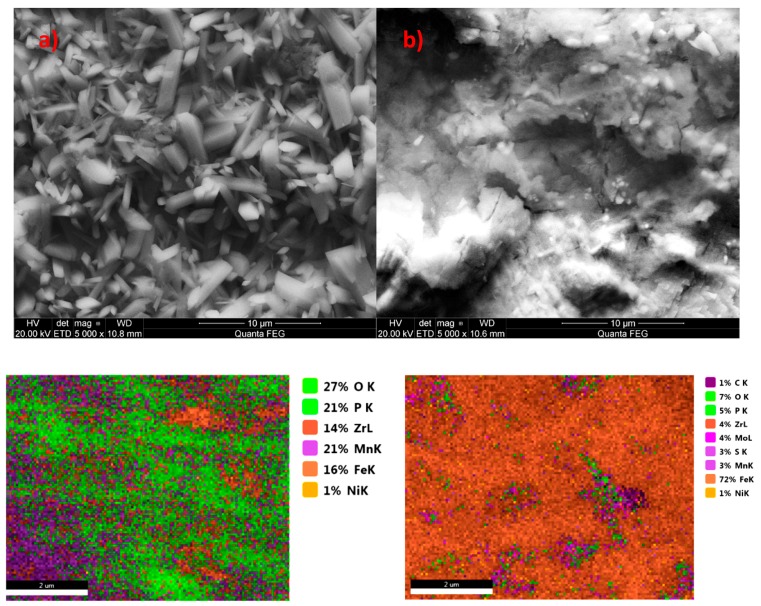
EDS and SEM photo for samples without addition of sodium molybdate (**a**) and for sample modified with sodium molybdate (**b**).

**Figure 5 materials-13-01416-f005:**
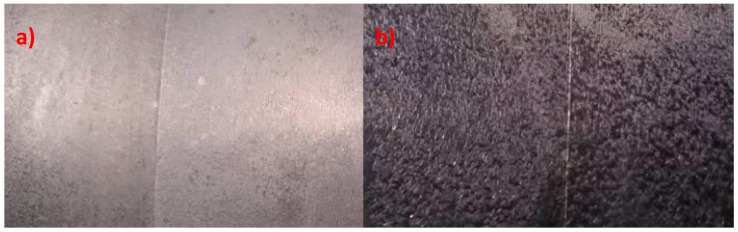
Effects of the substrate on the quality of the produced phosphate coating: (**a**) 235JR steel without chromium and molybdenum; (**b**) 42CrMoS4 + QT chromium-molybdenum steel.

**Figure 6 materials-13-01416-f006:**
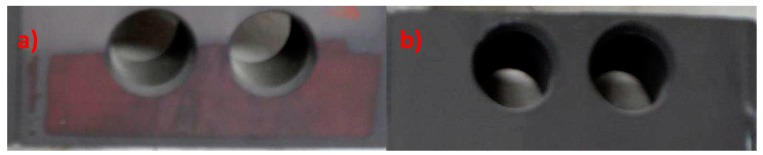
Samples produced under industrial conditions: (**a**) from the phosphating bath with an addition of molybdenum compound; (**b**) in a bath without the molybdenum compound. The samples were produced under the same temperature conditions.

**Figure 7 materials-13-01416-f007:**
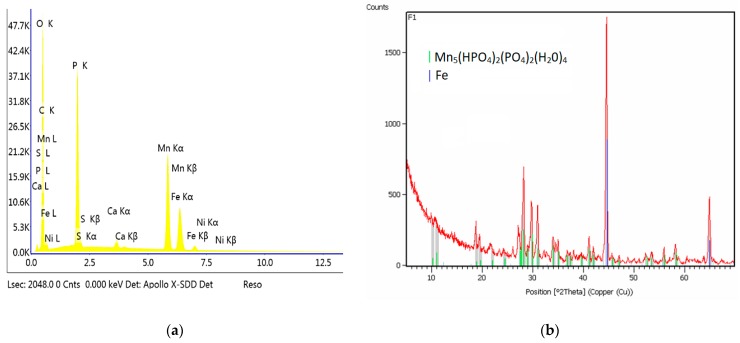
EDS (**a**) and X-ray diffraction (XRD) spectra (**b**) of the sample modified with calcium.

**Figure 8 materials-13-01416-f008:**
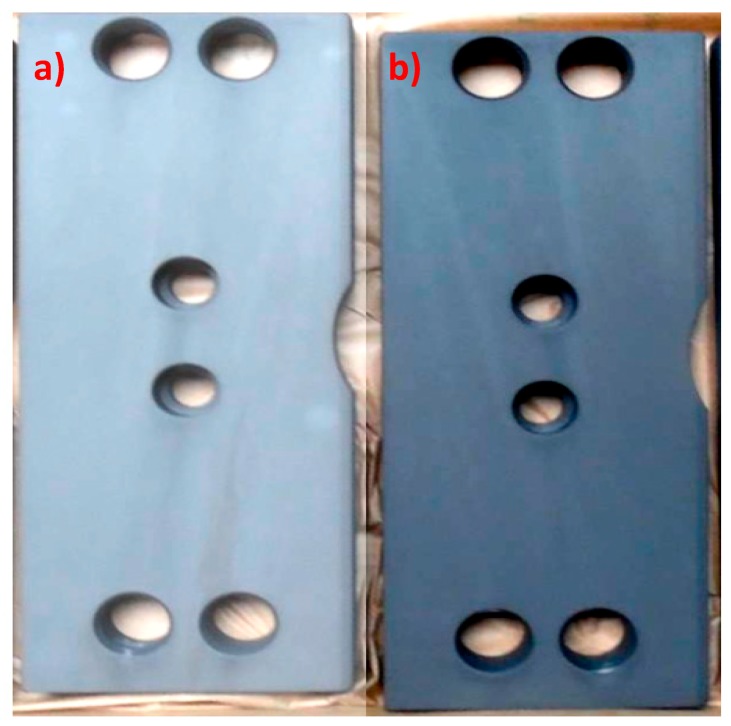
View of the sample covered with a manganese phosphate coating with calcium (**a**) and view of the sample covered with manganese phosphate without calcium (**b**). The samples were produced under the same conditions.

**Figure 9 materials-13-01416-f009:**
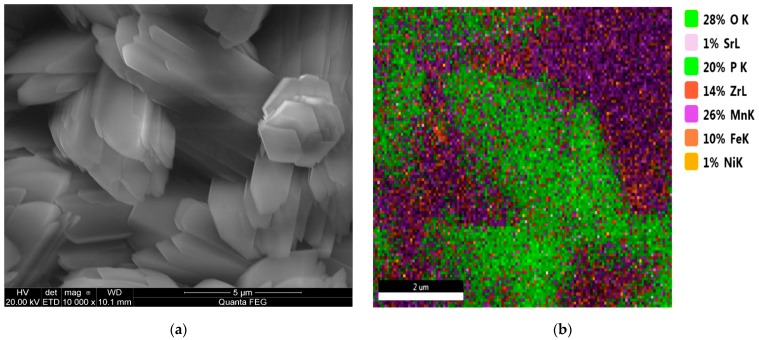
Coating produced from a phosphating bath containing strontium nitrate (**a**) and EDS image (**b**).

**Figure 10 materials-13-01416-f010:**
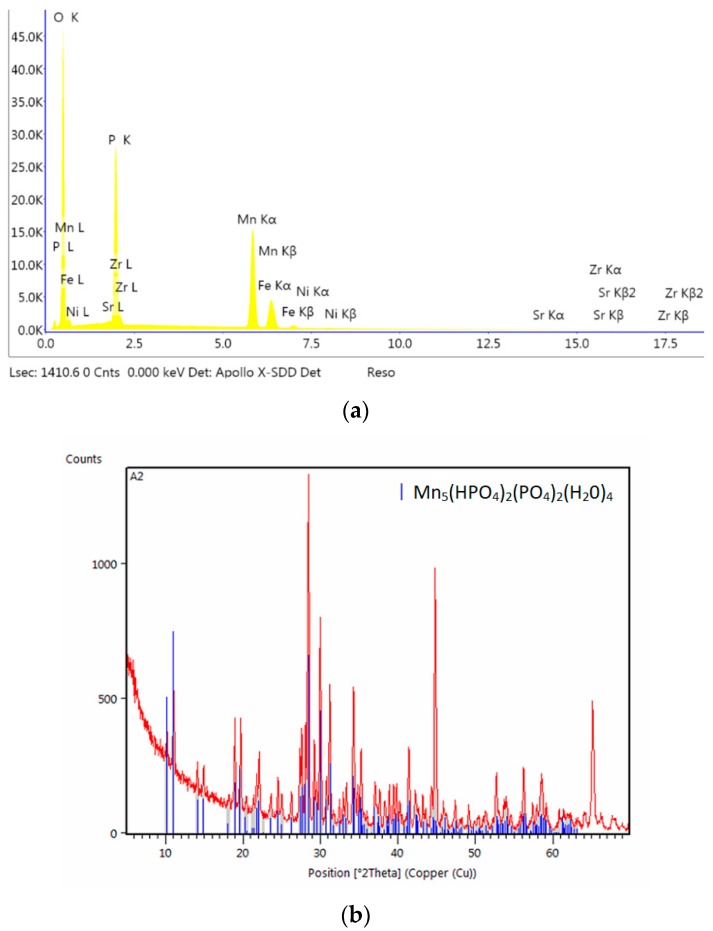
EDS (**a**) and XRD spectra (**b**) of the sample modified with strontium.

**Figure 11 materials-13-01416-f011:**
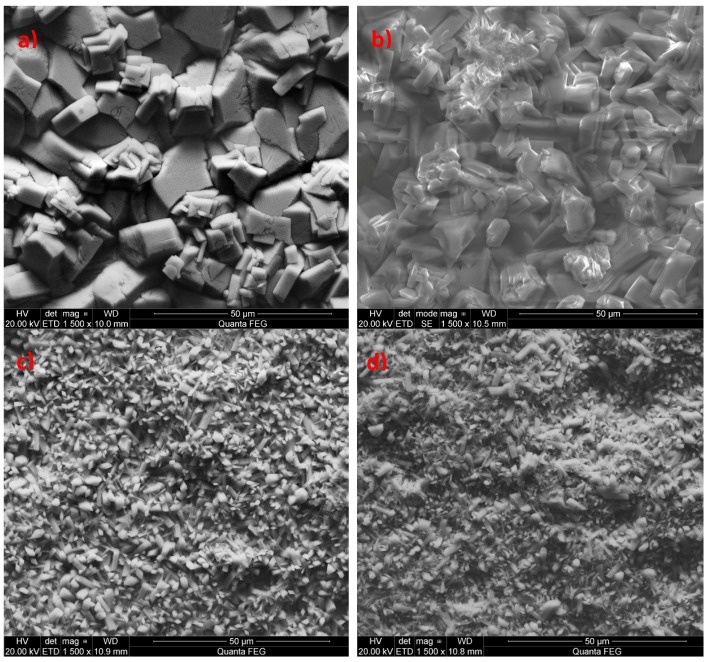
SEM image of phosphate coatings: (**a**) zinc phosphate coating, (**b**) manganese phosphatecoating modified with zinc, (**c**) manganese phosphate coating created under industrial conditions and passivated with hexafluorozirconic acid, (**d**) manganese phosphate coating produced under laboratory conditions and passivated with zirconyl oxychloride.

**Figure 12 materials-13-01416-f012:**
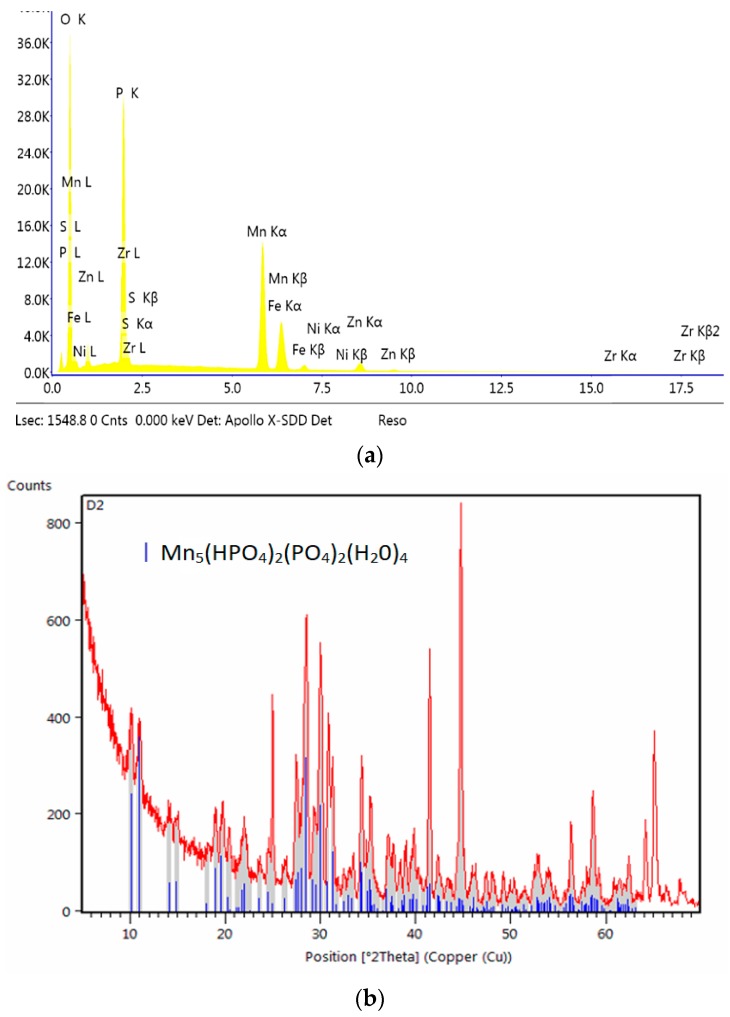
EDS (**a**) and XRD spectra (**b**) of the sample modified with zinc.

**Figure 13 materials-13-01416-f013:**
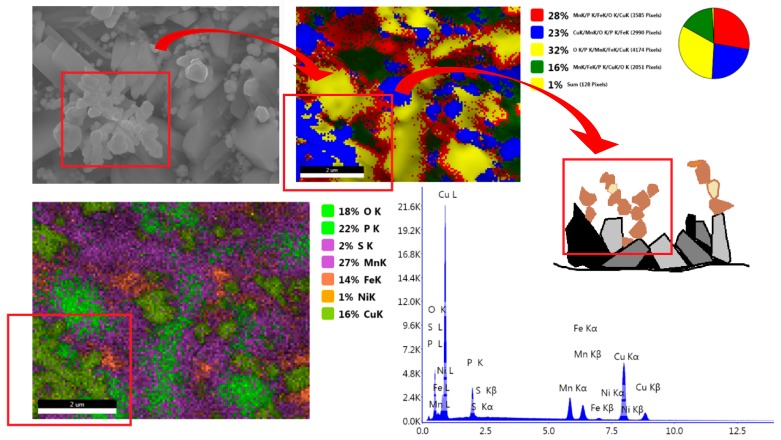
Phosphate coating modified with copper.

**Figure 14 materials-13-01416-f014:**
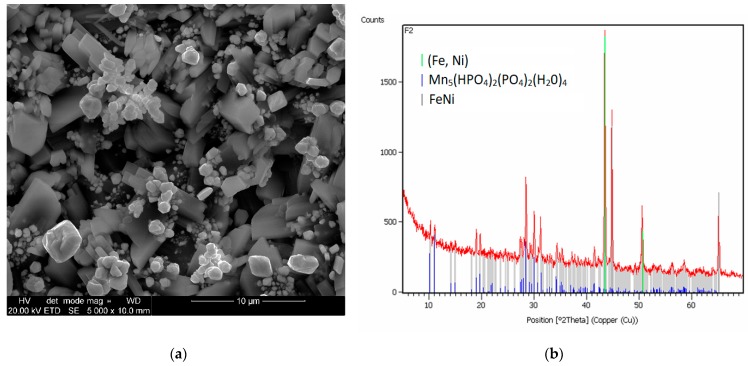
SEM photo (**a**) and XRD spectrum (**b**) for samples modified with copper nitrate.

**Figure 15 materials-13-01416-f015:**
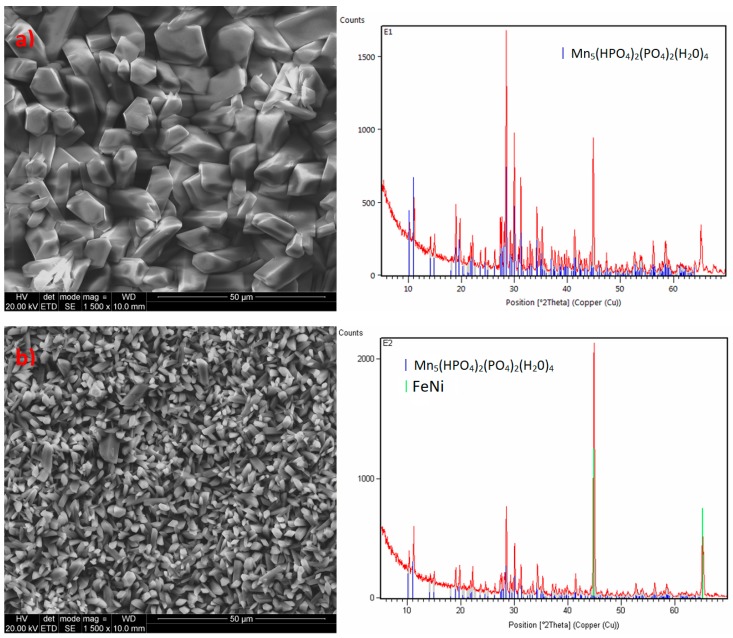
SEM photo and XRD pattern for samples modified with cadmium oxide (**a**) and samples modified with cadmium oxide and nickel compound (**b**).

**Figure 16 materials-13-01416-f016:**
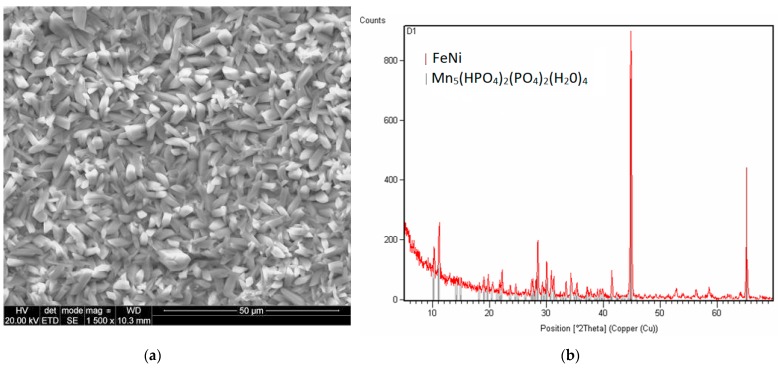
Microphotographs of the sample of a coating modified with barium nitrate (**a**) and XRD spectrum (**b**).

**Figure 17 materials-13-01416-f017:**
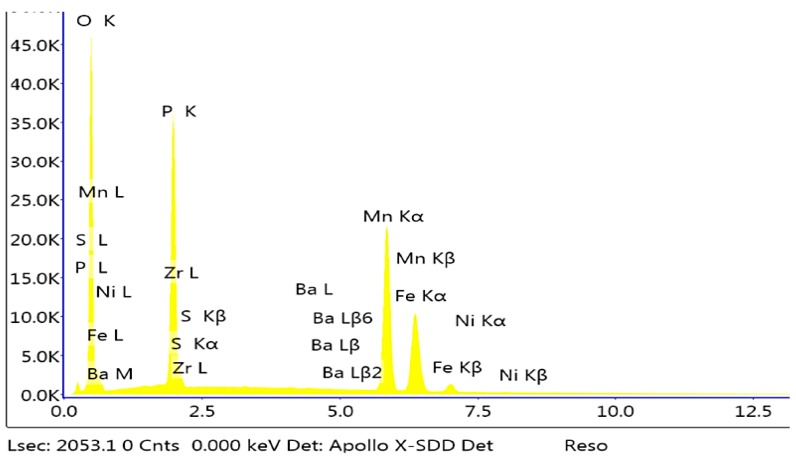
EDS analysis of a sample with a coating modified with barium nitrate.

**Figure 18 materials-13-01416-f018:**
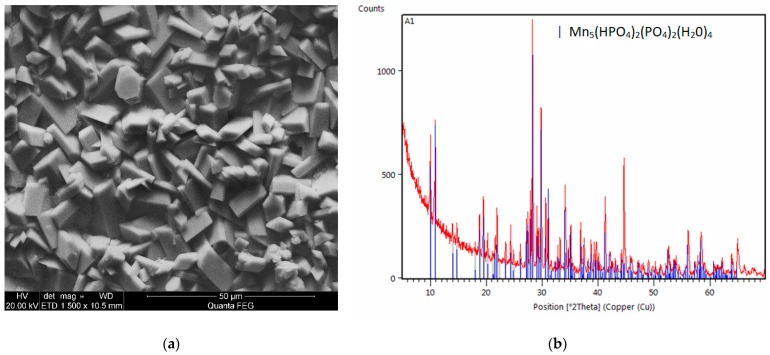
Microphotograph of the coating modified with cerium nitrate (**a**) and XRD spectrum (**b**).

**Figure 19 materials-13-01416-f019:**
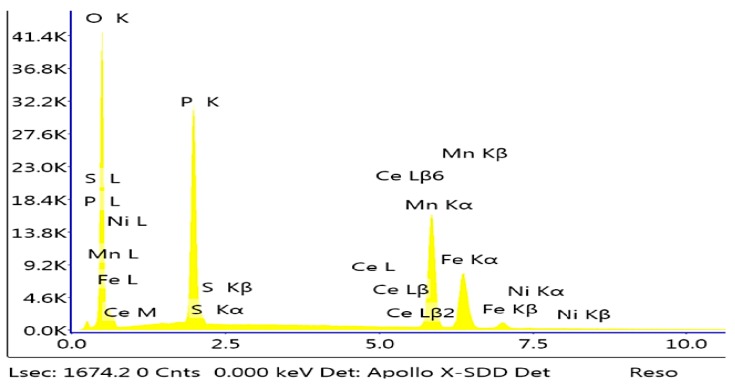
EDS spectrum of the sample modified with cerium.

**Figure 20 materials-13-01416-f020:**
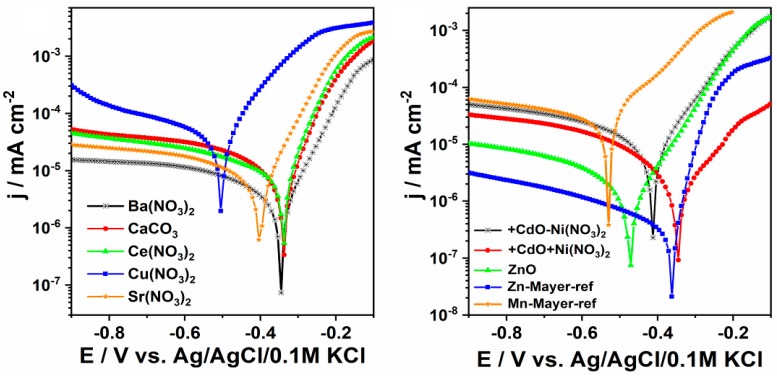
Electrochemical curves for the tested samples.

**Figure 21 materials-13-01416-f021:**
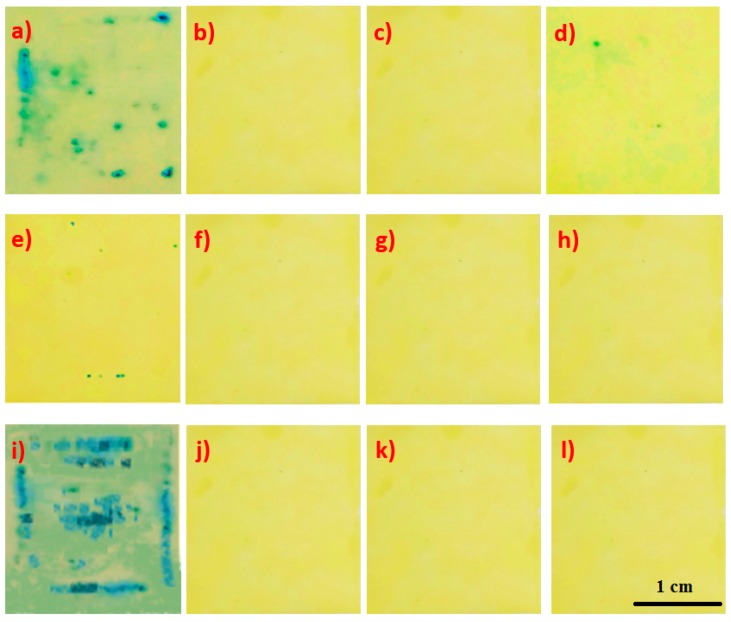
Analysis of the integrity of the tested samples: (**a**) sample modified with molybdenum, (**b**) sample modified with calcium, (**c**) sample modified with strontium, (**d**) sample modified with zinc, (**e**) sample modified with copper, (**f**) sample modified with cadmium, (**g**) sample modified with barium, (**h**) sample modified with cerium, (**i**) reference sample—pure metal without a conversion coating, (**j**) reference sample coated with the Mayr manganese coating, (**k**) reference sample with manganese coating additionally covered with Mayr mineral oil, and (**l**) reference sample covered with the Mayr zinc conversion coating.

**Table 1 materials-13-01416-t001:** Qualitative composition of steel subject to the phosphating process.

**Element**	**C**	**Si**	**Mn**	**P**	**S**	**Cr**	**Mo**	**Ni**	**Nb**	**Al**	**Cu**
**Weight Content (%)**	0.034	0.009	0.220	0.009	0.007	0.040	0.007	0.012	0.011	0.026	0.041
**Element**	**Co**	**B**	**Ti**	**W**	**Ca**	**Sn**	**Pb**	**Sb**	**Te**	**As**	**Fe**
**Weight Content (%)**	0.002	<0.0001	0.010	0.008	0.001	0.001	0.006	0.008	0.001	0.001	99.545

**Table 2 materials-13-01416-t002:** Quantitative composition of the proposed standard baths complete with stages [8].

Stage No.	Process Type	Composition of Solution	Process Conditions
1	Degreasing	10% (*m/m*) solution of NaOH 0.05% (*m/m*) sodium dodecyl sulfate (SLS)	mixing, temperature: 80–85 °C, 5 min.
2	Rinsing	Distilled water	mixing, room temperature, 3 min.
3	Etching	Solution of HCl 15% (*m/m*), 1–10 mL of 3% (*m/m*) H_2_O_2_, 0.05 g 1,2,3-benzotriazole	mixing, room temperature, 3 min.
4	Rinsing	Distilled water	mixing, room temperature, 1 min.
5	Activating	MnHPO_4_: 2 g Na_4_P_2_O_7_: 2 g H_2_O: 500 g	mixing, temperature: 40–45 °C, 4 min
6	Phosphating	H_3_PO_4_ (85%): 7.0 g Mn_3_(PO_4_)_2_: 15.0 g Mn(NO_3_)_2_: 6.0 g MnCO_3_: 0.5 g Ni(NO_3_)_2_: 0.3 g H_2_O: 531.5 g 1-methyl-3-nitroguanidine: 0.5 g	mixing, filtrated temperature: 95 °C, 15 min.
7	Rinsing	Distilled water	room temperature 3 min.
8	Passivation	ZrOCl_2_: 0.3 g Mn(NO_3_)_2_: 1.1 g NaNO_3_: 0.14 g HNO_3:_ 0.14 g CH_3_OH: 0.14 g Na_2_CO_3_: 5.4 g H_2_O: 100 g	mixing, 25–30 °C, 2 min.
9	Conservation with oil	Solution of mineral oil solution of mineral, emulsifying with water oil (ZWEZ 4999) produced by ZWEZ.	mixing, 75 °C, 2 min.

**Table 3 materials-13-01416-t003:** Chemical Composition of concentrated solution for manganese phosphate process [8].

Compound	Content (g)
H_3_PO_4_ (25% *m/m* solution)	40.0
Mn_3_(PO_4_)_2_	25.0
Mn(NO_3_)_2_	10.0
H_2_O	25.0
Ni(NO_3_)_2_	0.1
1-methyl-3-nitroguanidine (or nitroguanidine)	1.0

**Table 4 materials-13-01416-t004:** Chemical composition and working conditions for modified phosphate baths.

**Ba-Ni-Mn Solution**	**Zn-Ni-Mn Solution**	**Cd-Mn Solution**	**Cd-Ni-Mn Solution**	**Mo-Ni-Mn Solution**
H_3_PO_4_: 5.2 g Mn_3_(PO_4_)_2_: 3.25 g Mn(NO_3_)_2_: 1.3 g MnCO_3_: 0.5 g Ni(NO_3_)_2_: 0.1 g Fe: 0.2 g H_2_O: 125 g Ba(NO_3_)_2_: 1.0023 g 1-methyl-3-nitroguanidine: 0.3 g 95–98 °C, 15 min.	H_3_PO_4_: 5.2 g Mn_3_(PO_4_)_2_: 3.25 g Mn(NO_3_)_2_: 1.3 g MnCO_3_: 0.5 g Ni(NO_3_)_2_: 0.1 g Fe: 0.2 g H_2_O: 125 g ZnO: 0.30 g 1-methyl-3-nitroguanidine: 0.3 g 95–98 °C, 15 min.	H_3_PO_4_: 5.2 g Mn_3_(PO_4_)_2_: 3.25 g Mn(NO_3_)_2_: 1.3 g MnCO_3_: 0.5 g Fe: 0.2 g H_2_O: 125 g CdO: 0.10 g 1-methyl-3-nitroguanidine: 0.3 g 95–98 °C, 15 min.	H_3_PO_4_: 5.2 g Mn_3_(PO_4_)_2_: 3.25 g Mn(NO_3_)_2_: 1.3 g MnCO_3_: 0.5 g Ni(NO_3_)_2_: 0.1 g Fe: 0.2 g H_2_O: 125 g CdO: 0.10 g 1-methyl-3-nitroguanidine: 0.3 g 95–98 °C, 15 min.	H_3_PO_4_: 5.2 g Mn_3_(PO_4_)_2_: 3.25 g Mn(NO_3_)_2_: 1.3 g MnCO_3_: 0.5 g Ni(NO_3_)_2_: 0.1 g Fe: 0.2 g H_2_O: 125 g Na_2_MoO_4_: 0.30 g 1-methyl-3-nitroguanidine: 0.3 g 95–98 °C, 15 min.
**Cu-Ni-Mn Solution**	**Ce-Ni-Mn Solution**	**Sr-Ni-Mn Solution**	**Ca-Ni-Mn Solution**	**Standard Bath Composition**
H_3_PO_4_: 5.2 g Mn_3_(PO_4_)_2_: 3.25 g Mn(NO_3_)_2_: 1.3 g MnCO_3_: 0.5 g Ni(NO_3_)_2_: 0.1 g Fe: 0.2 g H_2_O: 125 g Cu(NO_3_)_2_: 0.2 g 1-methyl-3-nitroguanidine: 0.3 g 95–98 °C, 15 min.	H_3_PO_4_: 5.2 g Mn_3_(PO_4_)_2_: 3.25 g Mn(NO_3_)_2_: 1.3 g MnCO_3_: 0.5 g Ni(NO_3_)_2_: 0.1 g Fe: 0.2 g H_2_O: 125 g Ce(NO_3_)_2_: 1.0 g 1-methyl-3-nitroguanidine: 0.3 g 95–98 °C, 15 min.	H_3_PO_4_: 5.2 g Mn_3_(PO_4_)_2_: 3.25 g Mn(NO_3_)_2_: 1.3 g MnCO_3_: 0.5 g Ni(NO_3_)_2_: 0.1 g Fe: 0.2 g H_2_O: 125 g Sr(NO_3_)_2_: 2.5014 g 1-methyl-3-nitroguanidine: 0.3 g 95–98 °C, 15 min.	H_3_PO_4_: 5.2 g Mn_3_(PO_4_)_2_: 3.25 g Mn(NO_3_)_2_: 1.3 g MnCO_3_: 0.5 g Ni(NO_3_)_2_: 0.1 g Fe: 0.2 g H_2_O: 125 g CaCO_3_: 0.10 g 1-methyl-3-nitroguanidine: 0.3 g 95–98 °C, 15 min.	H_3_PO_4_: 5.2 g Mn_3_(PO_4_)_2_: 3.25 g Mn(NO_3_)_2_: 1.3 g MnCO_3_: 0.5 g Ni(NO_3_)_2_: 0.1 g Fe: 0.2 g H_2_O: 125 g 1-methyl-3-nitroguanidine: 0.3 g 95–98 °C, 15 min.

**Table 5 materials-13-01416-t005:** The crystallinity of the obtained coatings.

Modifying Substance	Average Crystal Size (µm)	Structure
Ba(NO_3_)_2_	<5	Large crystallites with oval edges
CaCO_3_	<5	Crystallites have the shape of sharp-tipped prisms
Ce(NO_3_)_2_	<25	Prismatic crystallites with distinct edges
Cu(NO_3_)_2_	<10	Spherical copper crystallites coated on manganese phosphate crystals
Na_2_MoO_4_	N/A	Not uniform structure, made up of several superimposed layers
Sr(NO_3_)_2_	<10	Multilayer lamellar crystallites with a regular, prism-like structure
CdO − Ni(NO_3_)_2_	<10	Crystals in the form of sharp-tipped thick prisms
CdO + Ni(NO_3_)_2_	10–15 (1–5 for aggregates)	Crystals in the form of sharp-tipped thick prisms with oval shapes; in some points, there are aggregates of small needles and ball-shaped crystals
ZnO	<20	Crystals have irregular shapes with large voids between them
Zn-reference	<10	Crystals in the form of irregular polyhedra
Mn-reference	<5	Crystals in the form of needles

**Table 6 materials-13-01416-t006:** Comparison of the qualitative composition of 42CrMoS4 + QT and S235JR steels [23,24].

**Stal 42CrMoS4 + QT**							
**C**	**Si**	**Mn**	**P**	**S**	**Cr**	**Mo**	**Ni**
0.38–0.45	max 0.40	0.6–0.9	max 0.025	0.02–0.04	0.9–1.2	0.15–0.30	max 0.40
**Stal 235JR**							
max 0.17	-	max 1.40	max 0.035	max 0.035	-	-	-

**Table 7 materials-13-01416-t007:** Values of the corrosion potential, current density, and corrosion rate for the modified samples.

Modyfying Substance	Corrosion Potential (Ecor/V)	Current Density (jcor/μAcm^−2^)	Corrosion Rate (CR/mpy)
Ba(NO_3_)_2_	−0.344	1.88	0.87
CaCO_3_	−0.337	3.50	1.62
Ce(NO_3_)_2_	−0.343	4.84	2.24
Cu(NO_3_)_2_	−0.504	47.4	21.90
Sr(NO_3_)_2_	−0.410	4.72	2.18
CdO − Ni(NO_3_)_2_	−0.418	4.96	2.29
CdO + Ni(NO_3_)_2_	−0.341	1.30	0.60
ZnO	−0.474	0.811	0.37
Zn-reference	−0.365	0.251	0.11
Mn-reference	−0.536	16.4	7.58

**Table 8 materials-13-01416-t008:** Results of coating mass measurement.

Modifying Substance	Sample Mass before the Measurement (g)	Sample Mass after Etching (g)	Coating Mass (g/m^2^)
Ba(NO_3_)_2_	2.4879	2.4637	0.03025
CaCO_3_	2.6173	2.6044	0.016125
Ce(NO_3_)_2_	2.8642	2.8538	0.013
Cu(NO_3_)_2_	2.9664	2.9446	0.02725
Sr(NO_3_)_2_	2.4373	2.4288	0.010625
CdO − Ni(NO_3_)_2_	2.1588	2.1503	0.010625
CdO + Ni(NO_3_)_2_	2.5747	2.5641	0.01325
ZnO	2.4288	2.4169	0.014875
Zn-reference	2.2019	2.1953	0.00825
Mn-reference	2.5920	2.5834	0.01075
Mn-reference with mineral oil	2.7351	2.7220	0.016375

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
