# Peer review of "Modified Manganese Phosphate Conversion Coating on Low-Carbon Steel"

_materials, 2020, doi:10.3390/ma13061416_

Round 1

Reviewer 1 Report

This is a useful an interesting study regarding the coating protection of carbon steel.

English language needs minor revision throughout the manuscript.

Title: “Phosphating Coating”: could it be changed to “Phosphate conversion coating”?

Determine the crystallinity of the obtained coatings, and discuss it comparatively with the type of additive. Mark on the diffractograms the potential existent phases.

Measurement units in Table 8: put them in US format, with “.” as decimal places.

Author Response

We send repiles to the reviewer's comments.

Reviewer 2 Report

The work describes the effect of metallic cations on the quality of manganese phosphate conversion coatings. In my opinion the manuscript is more a report than a scientific work as the authors limit to state their observations and do not confront them with data form the literature or advance any explanation for their findings.  

Anyway, the manuscript is well written and organized and present a large amount of work and, for that reason, I can give the manuscript another opportunity is these comments are addressed. Moreover, the following points should be revised.

  • The introduction section needs revision. The long list of types of coatings adds nothing and does not relate to the aim of the work. It would be more interesting to point the advantages of manganese phosphate pre- treatments vs. for instance phosphate coatings. Also, the actual state of the art regarding such treatments should be addressed.
  • Check lines 65 and 72 - it is not clear if the samples were degreased after or before immersion in toluene
  • Check equation 3 - something is missing or the CR result would be in mil/A.
  • In corrosion testing, the anodic and cathodic branches of the polarization curves should be determined separately. When the potential is scanned for approximately -1.0V in the positive direction the reactions occurring in the cathodic regime may alter the anodic behaviour or even the coating integrate (eg. H2 evolution). The results can, therefore, be compromised.
  • Check Figures numbers in lines 296, 297, 381
  • What do the authors mean by “The results of the electrochemical test (Fig. 20) show that copper is a better modifier, but the determined corrosion rate of 21.9 [CR/mpy] dos does not support this conclusion (Table 7)” (line 395). The CR is obtained from the electrochemical tests. Also, in point 5 of the conclusions “The corrosion rate for the 470 standard manganese sample is 7.58 CR/mpy, and for the manganese sample modified with zinc 471 – 0.37 CR/mpy. Current density is also highly variable – for the standard manganese sample, it is 16.4 [jcor/μAcm-2 ], and for the manganese sample modified with zinc – 0.811 [jcor/μAcm-2 472 ]”. There is no need to mention both CR and jcorr as. In practice, they are proportional and give the same information.
  • Finally, as phosphate conversion coatings are mainly used as pre-treatments (before organic coatings) an important aspect that should be mentioned is if the degree of adhesion to paints.

Author Response

We send answers to the reviewer's comment

Reviewer 3 Report

The number of conclusions should be reduced.  Some are not necessary, focus on the headline findings. 

The linear Tafel segments of the cathodic and anodic curves are quite difficult to evaluate, how many data sets did you use to determine the Jcorr?

Some indication as to the origin of Eq 3 would help the reader. 

Author Response

We send answers to the reviewer's comments.

Round 2

Reviewer 2 Report

Regarding the revised manuscript I think almost all comment were addressed although some deep English check is mandatory (namely n the new paragraphs).

However, the author's reply to comment 4 and the change made to the document:
   “Linear voltammetry (VE) scans were recorded from anode      to cathode potential”,
if true, it is completely wrong and, consequently, the results obtained are not reliable. In fact, if the reactions that occur during the cathodic branch can induce changes in the global behaviour of the material (as mentioned in comment 4), a polarization curve starting from the anodic limit is just absurd, as, at high potentials, corrosion occurs.

In my opinion, this is a very serious flaw in the experimental work, compromising all the conclusions and, therefore, I think that the work cannot be published in its current form.

Author Response

We thank the Reviewer for this valuable suggestion. All text was carefully checked and corrected. The relevant sections are highlighted blue and green in the revised version of the manuscript.

Round 3

Reviewer 2 Report

If, in fact, I can disregard the authors reply to my previous comment #4

“We are grateful for this comment and we apologize for the incorrect information provided in the experimental section regarding the run of linear voltammetry scan that probably occurs due to some accidental copy of the experimental protocol from our other reports. We did our best to perform the measurements and interpret data with due diligence and LV were recorded from the positive toward negative potential regime and thus any reactions occurring in the cathodic regime do not affect the anodic behaviour. Following our explanation, we correct the sentence in the experimental section where the direction of LV curve was described.”

that, of course, does not make any sense in corrosion testing, I can consider accepting the paper, according to the new corrections and those already made before. However, although several papers don´t use such experimental procedure, I must emphasize that in linear polarization measurements the anodic and cathodic curves should be carried out separately i.e. anodic curve should start a few mV below the OCP and the potential swept in the positive direction, while the cathodic branch should start a few mV above OCP and the potential scanned in the negative direction. This is the only procedure that guarantees that the “true” behaviour is obtained i.e. a behaviour that is not influenced be process occurring previously.

Regarding eq. 3 the introduction of the units in the conversion factor – absent in the first version - make now the equation dimensionally correct. This is a scientific paper so those inaccuracies should be avoided.